# Characterization of Electrospinning Chitosan Nanofibers Used for Wound Dressing

**DOI:** 10.3390/polym16141984

**Published:** 2024-07-11

**Authors:** Shahla H. Ali, Manaf A. Mahammed, Suhad A. Yasin

**Affiliations:** 1College of Medicine, University of Duhok, Duhok 42001, Iraq; shahla.hadi@uod.ac; 2College of Science, University of Duhok, Duhok 42001, Iraq; suhad.yasin@uod.ac

**Keywords:** electrospinning, nanofiber, chitosan, wound dressing, antibacterial

## Abstract

Wound dressings play a crucial role in promoting wound healing by providing a protective barrier against infections and facilitating tissue regeneration. Electrospun nanofibers have emerged as promising materials for wound dressing applications due to their high surface area, porosity, and resemblance to the extracellular matrix. In this study, chitosan, a biocompatible and biodegradable polymer, was electrospun into nanofibers for potential use in wound dressing. The chitosan nanofibers were characterized by using various analytical techniques to assess their morphology and biocompatibility. Scanning electron microscopy (SEM) revealed the formation of uniform and bead-free nanofibers with diameters ranging from tens to hundreds of nanometers. Structural analysis, including Fourier transform infrared (FTIR) spectroscopy and X-ray diffraction (XRD), elucidated the chemical composition and crystalline structure of the nanofibers. Furthermore, in vitro studies evaluated the cytocompatibility of the chitosan nanofibers with human dermal fibroblasts, demonstrating cell viability and proliferation on the nanofibers. Additionally, antibacterial properties were assessed to evaluate the potential of chitosan nanofibers in preventing wound infections. Overall, the characterization results highlight the promising attributes of electrospun chitosan nanofibers as wound dressings, paving the way for further investigation and development in the field of advanced wound care. This study has been carried out for the first time in our region and has assessed the antibacterial properties of electrospun chitosan nanofiber material. The created mat has shown efficaciousness against bacteria that are both gram-positive and gram-negative.

## 1. Introduction

Nanofiber materials have attracted significant attention due to their remarkable advantages in various applications. The interaction between these materials and their environment occurs primarily on the surface, so an increased surface area significantly enhances performance [1]. Research has shown that nanofibers have a much larger surface area compared with bulk or 2D materials [1,2]. Despite this, they retain the inherent physical properties of their base materials, which is crucial for applications demanding specific physical strength [3]. Nanofibers, ranging from a few nanometers to several micrometers, exhibit high surface area-to-volume ratios, small pore diameters, high porosity, low density, and superior mechanical properties, making them ideal for various applications [4]. They can be produced through methods like template synthesis, drawing, self-assembly, electrospinning, and phase separation [5]. Electrospinning is a highly effective method for producing submicron-sized fibers, with some as thin as several nanometers [6]. In this technique, an electric field applied between a needle capillary and a collector induces a surface charge on a polymer fluid, deforming it into a conical shape. When the electric field exceeds a certain threshold, electrostatic repulsion overcomes surface tension, ejecting a charged fluid jet from the Taylor cone tip. This charge density interacts with the external field, creating an instability that enhances a whipping mode, which stretches the polymer fibers and rapidly evaporates the solvent. Key parameters in electrospinning include polymer and solution properties like molecular weight, viscosity, conductivity, and surface tension, along with electrospinning conditions such as applied high voltage, tip-to-target distance, and flow rate [7]. One of the key advantages of electrospinning is its ability to fabricate nanofibers from a wide range of materials including synthetic polymers, natural polymers, and their blends. This versatility allows for the production of nanofibers with tailored properties suitable for various applications such as tissue engineering, drug delivery, filtration, and wound dressing [8]. Chitosan is an N-deacetylated product of chitin, the second-most abundant natural polysaccharide next to cellulose, which is embedded in a protein matrix of a crustacean shell or a squid pen [9].

Chitosan has garnered significant attention in wound healing due to its biocompatibility, biodegradability, and antimicrobial properties [10] Wound healing is a critical biological process integral to tissue regeneration and repair [11]. Traditional wound dressings often fail to maintain moisture or provide antibacterial protection. For instance, gauze offers limited protection due to poor hydrophilicity [12,13], while hydrogel dressings soothe wounds but risk maceration and bacterial growth [14]. Alginate dressings accelerate healing but lack bacteriostatic effects [15], and collagen dressings promote healing but may cause immune rejection [16]. Silver nanoparticles dressings are antibacterial but pose health risks due to accumulation [17]. To enhance wound healing, dressings need to maintain moisture and offer antibacterial properties. Electrospun nanofiber dressings excel in promoting tissue regeneration and wound healing by enhancing cell proliferation and migration [18]. Chitosan-based dressings, being biocompatible, are especially effective [19]. Thus, electrospun nanofibers represent a promising advancement in wound care.

The synergy between chitosan and electrospinning presents a compelling strategy for the development of advanced wound dressings. Chitosan-based nanofibers offer several advantages, including high surface area-to-volume ratio, tunable mechanical properties, and excellent biocompatibility, which are essential for promoting wound healing. Additionally, the antibacterial properties of chitosan make it particularly suitable for combating infections in chronic wounds [20].

Chitosan-based nanofibers offer several advantages for wound healing. Their high surface area-to-volume ratio provides an extensive contact area with the wound bed, facilitating cell attachment, migration, and proliferation. The porous structure of these nanofibers enables efficient moisture management and creation of a conducive environment for wound healing by maintaining optimal moisture levels [21].

In summary, electrospun chitosan nanofibers hold great promise for wound dressing applications, offering a combination of biocompatibility, antimicrobial activity, and mechanical properties essential for effective wound healing [22]. Porosity and wettability are critical characteristics of chitosan nanofibers that significantly influence their performance in various applications, particularly in fields such as tissue engineering, drug delivery, and wound dressing. Wettability refers to the ability of the surface of a material to interact with a liquid, determining whether the liquid spreads or beads up on the surface [23].

Chitosan nanofibers typically exhibit hydrophilic properties, which means that they have a high affinity for water. Additionally, the hydrophilic surface of chitosan nanofibers enhances the absorption and retention of wound exudate in wound dressing applications, maintaining a moist wound environment conducive to healing. In summary, the porosity and wettability of chitosan nanofibers play integral roles in their performance and functionality across various applications. Their high porosity facilitates fluid transport and molecular diffusion, while their hydrophilic nature promotes cell adhesion and wound healing processes, making them versatile materials for biomedical and biotechnological applications [24].

Antibacterial activity is highlighted for its significant impact on public health, hygiene, and safety across various domains including healthcare, the food industry, and environmental protection. Antibacterial agents play a crucial role in preventing infections, promoting wound healing, curbing healthcare-associated infections, preserving food quality, and combating antibiotic resistance. Notably, various infectious bacterial species, including both gram-positive (e.g., Staphylococcus aureus) and gram-negative (e.g., Pseudomonas aeruginosa), frequently infiltrate wound areas, hindering the healing process and necessitating effective antibacterial strategies [25]. Because of its remarkable antibacterial qualities—which are a result of interactions with bacterial membranes and the prolonged release of active chitosan molecules—chitosan nanofibers have become more and more popular in biomedical applications [26]. Because of their larger surface area and improved interaction with bacterial cells, these nanofibers are more effective against bacteria than bulk chitosan, which makes them useful for antimicrobial coatings, tissue scaffolds, and wound dressings [27]. Chitosan nanofiber functionalization minimizes systemic effects by encasing antibacterial compounds for targeted distribution [28]. The study emphasizes how adaptable chitosan nanofibers are as strong antibacterial substances to fight illnesses and improve public health [29]. Trifluoroacetic acid (TFA) provides exact control over the morphology and content of chitosan nanofibers electrospun by [30]. TFA-based electrospinning yields homogeneous nanofibers with a fine structure and advantageous mechanical properties. Due to their high surface area-to-volume ratio, biocompatibility, and ability to promote cellular functions, these nanofibers show tremendous promise in biomedical applications such as tissue engineering, wound healing, and drug delivery [31].

The knowledgeable gap in the study of nanofibers used for wound dressing lies in the lack of the comprehensive understanding of their properties, particularly in specific geographical regions or contexts. While there exists research on chitosan-based wound dressings and electrospun nanofibers, there may be limited studies focusing on the detailed characterization of these materials fitted for wound dressing applications in certain regions.

This study focuses on the characterization of chitosan nanofibers for wound dressing applications, particularly within the region of interest, aiming to contribute to wound care technology advancement and innovative wound dressing development tailored to local patient needs. The research outlined in the provided material delves into a systematic exploration of how different operational parameters influence the morphology, wettability and porosity of electrospun chitosan nanofibers. Detailed analysis is conducted on parameters including applied high voltage, solution flow rate, solution concentration, needle diameter, and tip-to-target distance. The primary focus of the current study is to evaluate the appropriateness of these nanofibers for showcasing antibacterial properties. More specifically, the research aims to assess their efficacy against both gram-positive and gram-negative microorganisms.

## 2. Materials and Methods

### 2.1. Materials

Chitosan polymer (300–1000 cps) was obtained from Glentham Life Sciences Ltd., Corsham, UK, Trifluoroacetic acid (TFA), with a purity of ≥99.9%, and the salt tetrabutylammonium bromide (TBAB), with a purity of ≥99% and a molecular weight of 322.4 g/mol, were provided by CARL ROTH, Karlsruhe, Germany

### 2.2. Determination of the Molecular Weight

To determine the average of molecular weight of the chitosan used in this work, we employed the intrinsic viscosity method using an Ostwald viscometer at 25 °C. A buffer solution composed of 0.15 M ammonium acetate and 0.2 M acetic acid was prepared. To estimate the average molecular weight of the chitosan, the Mark–Houwink equation [*η*] = k.M^a^ was used, where [*η*] is the intrinsic viscosity, M the viscosity-average molecular weight, and k= 9.66 × 10^−5^ dm^3^/g and a = 0.742 and both are empirical constants [32]. The calculated viscosity-average molecular weight of the chitosan used in this work was found to be M = 418 kDa.

### 2.3. Determination of Degree of Deacetylation (DD)

To determine the degree of deacetylation (DD) of the chitosan sample, we used Fourier transform infrared (FTIR) spectroscopy transmission data with the formula [33], [DD% = 100% − ((A_1320_/A_1420_ − 0.3822) / 0.03133)%]. A1320/A1420 is the absorbance ratio of the band at 1320 cm^−1^ for N-acetylglucosamine to the reference peak at 1420 cm^−1^, which was suggested by [34]. This method aligns with proton nuclear magnetic resonance (¹H NMR) results, is unaffected by humidity and water content, and provides low experimental error [33]. The determined value of DD was 89%.

### 2.4. The Preparation of Electrospinning Solutions

The preparation of 2%, 6%, and 8% chitosan solutions involved dissolving chitosan in trifluoroacetic acid (TFA) and sonicating each solution separately for 3 h at 55 °C using an ultrasonic cleaner (Model: VGA-1620 QTD) Gemini Lab Apeldoorn, The Netherlands. The electrospinning process utilized stainless steel needles with sizes ranging from 0.3 to 0.6 mm, with the solutions loaded into 5 mL plastic syringes.

### 2.5. Electrospinning Process

The electrospinning system that was used in this study is shown in Figure 1. It consists of three main parts, namely, the syringe pump type (ZS-100) Chonry, Baoding, China, 0–40 KV high voltage power supply Precision Pump Co., Ltd. Baoding, China, and a plate collector covered by aluminum foil that was used as a target to collect electrospun non-woven nanofibers. The target was enclosed within an airtight plastic chamber with glass windows. The syringe pump was put outside of the chamber with its needle leaded through the glass window into the chamber towards the plate collector. Throughout this study, the distance between the needle tip and the collection was varied from 120 to 180 mm. Flow rates ranged from 0.1 to 0.5 mL/h, and high voltages between 14 and 20 kV were applied. Subsequently, the mats were extracted from the aluminum foil target post-electrospinning and transferred to a desiccator for subsequent analysis.

### 2.6. Antibacterial Activity Method

Chitosan samples, comprising nanofibers and polymers, were manufactured according to the proper procedures and aseptically in order to assess the antibacterial activity against *E. coli* and *S. aureus*. The standard disc diffusion method was then used to test the samples. Standard antibiotic discs were placed on the agar surface after bacterial suspensions at uniform quantities were inoculated onto Mullar Hinton agar plates, manufactured by Muller Hinton Agar, Difco, Detroit, MI, USA For the polymer samples, chitosan nanofibers were directly deposited after 24 h of incubation at 37 °C. The inhibition zones surrounding the chitosan samples were assessed in terms of hours for bacterial growth to determine whether there were any appreciable variations in the antibacterial activity of the polymer and chitosan nanofiber samples against the two bacterial strains. The widths of these zones were measured and statistically analysis was performed.

### 2.7. Characterizations

#### 2.7.1. Scanning Electron Microscope

The surface morphology of the chitosan nanofibers was investigated using an The SEM (scanning electron microscope) used is a Phenom Pro G6 model manufactured by Thermo Fisher Scientific, which is based in Waltham, MA, USA. Image J software Java 1.8.0_172 (64-bit) was utilized to analyze the diameters of the chitosan nanofibers and their distribution. To determine the mean diameter and standard deviation, approximately one-hundred fibers were randomly selected from various regions of an SEM image.

#### 2.7.2. Fourier Transform Infrared Spectroscopy (FTIR)Spectroscopy

The functional groups of the produced samples were recorded using a spectrophotometer (Model 1800) is manufactured by Shimadzu Corporation, based in Kyoto, Japan, operating at wavelengths ranging from 400 to 4000 cm^−1^ with 100 scans. Subsequently, Origin Pro 2021 Software was employed to plot the curve.

#### 2.7.3. X-ray Diffraction (XRD)

To explore the crystallinity characteristics, the 2θ angles of the samples were measured using a Philips diffractometer (model PW1730), Eindhoven, The Netherlands. Measurements were taken within the range of 0 to 70 degrees at a speed of 0.02 s^−1^, with the instrument operating at 35 kV and 25 mA.

## 3. Results and Discussion

### 3.1. Parameter Effects on Nanofiber Morphology

#### 3.1.1. Effect of High Voltage

The effect of high voltage on nanofiber diameter is a critical aspect in the electrospinning process. Generally, increasing the voltage leads to the elongation and thinning of the polymer jet ejected from the spinneret. This results in the formation of finer nanofibers with a reduced diameter. Higher voltages create stronger electrostatic forces, which cause the polymer solution to stretch and elongate more, leading to thinner fibers. As the high voltage varies, the diameter of produced nanofibers fluctuates while the extrusion rate remains constant. Increasing the high voltage from 14 to 16, 18, and 20 kV resulted in a decrease in nanofiber diameter. Figure 2 displays SEM micrographs of the produced nanofibers, along with corresponding histograms illustrating the variation in diameter distribution for each high voltage value. The decrease in nanofiber diameters can be attributed to changes in electrostatic forces induced by the higher voltage, causing the solution to extrude more rapidly from the needle. The voltage increase generates a stronger electric field, propelling the jet ejected from the nozzle further towards the collector, ultimately producing finer nanofibers. A similar trend was observed by [35]. in a previous study where they noted a slight decrease in nanofiber diameters with an increase in applied voltage at a fixed distance. Under constant distance and flow rate conditions, the average fiber diameter decreased from 910 nm at 5 kV to less than 150 nm at 25 kV [35].

#### 3.1.2. Effect of Flow Rate

Variations in the flow rate of the precursor solution passing through the needle have a notable impact on the morphology of electrospun nanofibers. As the flow rate increases (ranging from 0.1 to 0.5 mL/h), there is a corresponding increase in the mean nanofiber diameter, measured at 180, 184, 193, 198, and 209 nm, as illustrated in the SEM micrographs and histograms presented in Figure 3. This increase is attributed to the expansion of the initial radius and volume of the electrospinning jet. Achieving uniform, beadless electrospun nanofibers requires careful control of the precursor solution flow rate. Inadequate withdrawal of solution from the nozzle tip may lead to intermittent leaking and bead formation, as demonstrated by the observed impact of the flow rate on the electrospun fiber diameter. At flow rates below 0.2 mL/h, the spinning fluid dries up and the electrospinning process halts. Consequently, an increase in flow rate results in larger fiber diameters and a wider distribution of fiber sizes. This effect may lead to the formation of larger-diameter beaded fibers at higher flow rates due to a decrease in electrostatic density [36,37].

#### 3.1.3. Effect of Tip-to-Target Distance

The influence of the distance between the needle tip and the collector is depicted in Figure 4. As the distance increased from 12 cm to 14 cm, 16 cm, and 18 cm, the fiber size decreased, measuring 258 nm, 256 nm, 228 nm, and 179 nm, respectively.. The distance between the needle tip and the collector is influenced by various factors, including deposition time, evaporation rate, and the interval between whipping or instability. This distance determines the speed of the jet and the distance it travels before settling on the collector, allowing sufficient time for solvent evaporation [38,39,40,41].

#### 3.1.4. Effect of Needle Diameter

SEM micrographs of chitosan nanofibers produced with four different needle sizes are presented in Figure 5. The mean fiber diameters for samples a, b, c, and d were determined as 226, 218, 172, and 134 nm, respectively. This indicates a relationship between needle diameter and fiber diameter. The increase in fiber diameter with larger needle sizes can be attributed to the expanded size of the droplet at the needle tip. As the droplet size increases, the surface tension decreases, requiring less columbic force to initiate jet formation under the same applied voltage. Consequently, the jet accelerates more rapidly, giving the solution less time to stretch and elongate before collection, leading to larger fiber diameters. This finding is consistent with previous research [42], which observed that nanofibers produced with smaller needle diameters exhibited smoother, thinner, bead-free characteristics and showed no signs of agglomeration.

#### 3.1.5. The Effect of Solution Concentration

The findings reveal that nanofibers produced at different chitosan/TFA concentrations exhibit consistent length, smooth texture, and absence of beads. Figure 6 demonstrates the significant impact of polymer concentration on the average diameter of electrospun chitosan-based nanofibers. As the chitosan/TFA solution concentration increases from 2% to 6% and 8% by weight, the average fiber diameters noticeably increase, measuring 144, 172, and 367 nm, respectively. This trend can be attributed to the elevated concentration and viscosity of the polymeric solution, which impedes the stretching of the charged jet. Consequently, polymer chains elongate and become more entangled to overcome surface tension. The increased size and weight of the fibers result in denser mat formation during electrospinning [43,44].

#### 3.1.6. Effect of High Voltage with Salt on Nanofiber Diameter

When employing a 6% solution with 0.0003 gm of tetra butyl ammonium bromide (TBAB), the resultant nanofiber diameter varied with voltage changes while keeping the extrusion rate constant. With high voltage increased from 14, 16, and 18 to 20 kV, the diameter of the nanofiber decreased. SEM micrographs of the produced nanofibers are shown in Figure 7, together with matching histograms that show how the diameter distribution of the nanofibers varies for each high voltage value. A change in electrostatic forces caused by the higher high voltage is responsible for the decrease in nanofiber diameters because it causes the solution to extrude from the needle more quickly. The expelled jet travels farther from the nozzle to the collector due to the stronger electric field created by the voltage increase [45,46].

### 3.2. Water Contact Angle Measurement

In biomedical applications, the wettability of nanofibers is an essential feature that affects protein absorption and cell adhesion. The water contact angles of the nanofibers were then measured by carefully dropping distilled water onto the sample surface. A video monitor was used to measure contact angles, and several measurements were made for every sample group. In order to comprehend the wetting behavior of the films, the wettability of the samples was evaluated using the measurement of water contact angles. The water contact angles values 35.6, 40.8, and 73.4 were significantly increased with different concentrations of chitosan nanofibers (2, 6, and 8 wt%), as shown in Figure 8. On the other hand, the water contact angle values decreased with increasing the high voltage levels (14, 16, 18, and 20 kV), as shown in Figure 9 (97.0, 89.6, 78.3, and 73.4). The surface became hydrophilic, and Figure 10 (15.5, 16.4, 19.2, 20.1, and 22.2) and Figure 11 (17.2, 18.4, 19.4, and 23.3) illustrate this pattern with higher flow rates (0.1, 0.2, 0.3, 0.4, and 0.5 ml/h) and variable needle diameters (0.3, 0.4, 0.5, and 0.6 mm). These results imply that when the tip-to-target distance and high voltage are raised, the hydrophobic characteristics of chitosan nanofibers may be connected to the surface roughness that is produced [47,48].

### 3.3. Impact of Nanofiber Porosity

Every nanofiber membrane in our investigation has a porosity greater than 30%. A relationship between membrane porosity and nanofiber diameter wettability was found. Since the random deposition of nanofibers occurs during electrospinning to generate the non-woven nanofiber, the fibrous architecture should be related to the applied voltage, the material dielectric characteristic, and the nanofiber. Additionally, it was shown that, as predicted, a high porosity increased the permeability of the membrane to air and moisture. The fiber diameter, which is mostly influenced by the flow rate during the electrospinning process, has a significant impact on the porosity in the electrospun nanofiber [49]. By modifying the electrospinning parameters such as solution concentration, applied voltage, flow rate, tip-to-target distance, and all other solution and process factors it is possible to control the fiber diameter. Controlling the porosity percentage in contrast to fiber diameter is more challenging and closely correlated with the former [50].

The subsequent formulas were employed to calculate the porosity of the nanofiber mat [51]
Porosity (%) = [1 − Density_apparent_ (g/cm^3^)/Density_material_ (g/cm^3^)] * 100%(1)
Density_apparent_ (g/cm^3^) = Mass (g)/(Thickness (cm) * Area (cm^2^))(2)

The sample has a thickness of 0.05 mm, a mass of 0.0016 g, and an area of 1 cm², with a material density of 1 g/cm³. As observed in Figure 12, increasing the flow rate from 0.1, 0.2 to 0.3, 0.4, and 0.5 mL/h decreased the porosity. Figure 13 shows that the porosity increased with an increase in the high voltage, but with adding salt the porosity increased at 14,16,18 kV, then started to decrease after 18, 20, and 22 kV with salt (0.0003 g), while Figure 14 shows a decrease in porosity with an increase in the needle diameter from 0.2 to 0.5 mm, and then a raise in porosity when the needle diameter increased from 0.5 to 0.6 cm, and then stayed the same from 0.6 to 0.7 mm. In Figure 15, the porosity increased when the concentration increased from 2% to 6%, then decreased at 8%, respectively, comparable to [52].

### 3.4. Fourier Transform Infrared Spectroscopy (FTIR)

Figure 16 shows that the Fourier transform infrared (FTIR) spectra of pure chitosan and chitosan nanofibers reveal notable peaks indicative of their chemical compositions. In pure chitosan, peaks at 3460 cm^−1^ and 2873 cm^−1^ signify O-H stretching vibrations and C-H stretching vibrations in CH_2_ and CH_3_ groups, respectively. Peaks at 1647 cm^−1^ and 1577 cm^−1^ [53] correspond to the amide bands, specifically amide I and amide II, suggesting the presence of protein-like structures. The peaks at 1423 cm^−1^ and 1381 cm^−1^ are associated with CH_2_ and CH_3_ bending vibrations, while the peak at 1330 cm^−1^ represents amide III. Comparatively, the FTIR spectrum of chitosan nanofibers exhibits similar peaks, albeit with slight shifts and variations. Notably, the O-H stretching vibration peak shifts to a lower wavenumber (3464 cm^−1^), indicating potential alterations in hydrogen bonding interactions due to nanofiber morphology or structure. Despite these differences, common features such as C-H stretching vibrations, amide bands, and glycosidic bond skeletal vibrations persist in both spectra, signifying the preservation of the fundamental chitosan structure. These spectral distinctions offer insights into the chemical composition and structural modifications between pure chitosan and chitosan nanofibers.

In comparing the FTIR spectra peaks of pure chitosan and chitosan nanofibers, notable differences and similarities emerge, shedding light on the structural transformations induced by the electrospinning process. In pure chitosan, characteristic peaks are observed at 3460 cm^−1^ (O-H stretching), 2873 cm^−1^ (C-H stretching), 1647 cm^−1^ (amide I band), and 1577 cm^−1^ (amide II band), among others [54]. Conversely, chitosan nanofibers exhibit shifts or alterations in these peaks, with notable peaks observed at 3464 cm^−1^, 2981 cm^−1^, 1573 cm^−1^, and 1423 cm^−1^, respectively [55]. These changes in peak positions or intensities suggest modifications in the hydrogen bonding pattern, molecular conformation, and chemical environment of chitosan molecules post-electrospinning. Shifts towards lower wavenumbers in certain peaks may indicate increased interactions or structural rearrangements induced by the electrospinning process. Additionally, variations in peak intensities or shapes reflect alterations in functional groups or molecular configurations.

These spectral differences highlight the impact of processing techniques on the structural properties of chitosan nanofibers, offering valuable insights for their tailored design and optimization for various applications.

### 3.5. X-ray Diffraction (XRD)

The XRD pattern for both of chitosan polymer and electrospun chitosan nanofibers are depicted in Figure 17. The diffraction pattern of bulk chitosan shows a high-intensity broad hump at the angle 2θ = 20° and a relatively smaller and yet broader hump at 2θ = 29°. This indicates that the chitosan used in this work has somewhat low oriented planes of small crystallites at those angles. Furthermore, it has an amorphous structure. On the other hand, the XRD pattern of the non-woven nanofiber mat of chitosan shows that the two humps were diminished and disappeared. This implies that the electrospun nanofibers have a great effect on the crystalline structure of chitosan by decreasing the preferred orientation of reflection planes. This could be attributed to the compatibility of the electrospun chitosan nanofiber mat that causes the disruption of the chitosan crystalline network in the presence of nanofibers. The disappearing of XRD peaks in case of electrospun nanofibers can be interpreted by the fact that, during electrospinning, the solution jet is elongated by the effect of the electric field, and the solvent evaporates and nanofibers are solidified before chitosan polymer macromolecules are arranged into crystals since they need days for crystallization. Therefore, the chitosan nanofiber crystal structure is not observed within the XRD pattern [56].

### 3.6. Antibacterial Activity

The antibacterial activity of the produced nanofibers is shown in Figure 18 and Table 1. The chitosan natural polymer showed relatively no antibacterial activity against *E. coli* and *S. aureus*. On the other hand, results for the chitosan nanofibers revealed a noticeable antibacterial potential against both tested gram-negative (*E. coli*) and gram-positive (*S. aureus*) bacteria, as shown in Figure 18 and Table 1. Chitosan has been shown in numerous investigations to have varying antibacterial effects against both gram-positive and gram-negative bacteria. Many studies reported that chitosan showed variable antibacterial effect against gram-negative and gram-positive bacteria. Using a range of testing methods, multiple researchers have investigated the antibacterial activity of distinct chitosan molecules from different sources. Molecular weight (Mw), the degree of deacetylation (DD), pH, test strains, and other external and intrinsic variables were frequently the cause of differences in the results obtained. Conversely, disparities in the outcomes of chitosan’s antimicrobial susceptibility tests might also be caused by the use of different testing techniques [57,58]. In the current study, the disk diffusion method was used to measure the in vitro susceptibility of bacteria to chitosan, which is the most popular method used to examine the antimicrobial activity of natural products including chitosan. This method measures the inhibition zone size, which is then converted to categories of susceptible/intermediate/resistant based on CLSI recommendations [59]. Additionally, it has been reported that chitosan as a natural polymer (bulk chitosan) exhibits low-to-moderate antibacterial effectiveness compared with that of the chitosan nanofibers, which are well known to exhibit high, enhanced, and superior antibacterial effectiveness. This is mainly because chitosan nanofibers have a significantly higher surface area compared with bulk chitosan, which allows for more interactions with bacteria, potentially enhancing antibacterial activity. Further, due to their nano-scale dimensions, chitosan nanofibers can provide better contact efficiency with bacteria compared with bulk chitosan. This increased contact can lead to more effective antibacterial action [60,61,62]. These results provide insights into the efficacy of chitosan as an antibacterial agent against *E. coli* and *S. aureus*, and show its value in various biomedical and industrial applications. Furthermore, it may suggest that the topology of chitosan-based fibers presents a significant opportunity for creating flexible fibrous scaffolds for incorporating several antibacterial and bioactive substances into the chitosan-based structure of fibers to boost the antibacterial effects and prevent bacterial infections.

## 4. Conclusions

Electrospinning is a useful technique for utilizing biopolymeric ingredients to create non-woven mats for wound healing. Throughout all of this research, electrospun chitosan nanofibers were prepared successfully. In order to achieve the best fine, smooth, free-of-beads chitosan nanofibers to be useful for an antibacterial wound dressing, investigations of the effects of the applied high voltage, solution flow rate, solution concentration, the tip-to-target distance, and the needle diameter on the morphology, wettability, and porosity of the electrospun,, chitosan nanofibers were carried out. To summarize, the primary conclusions drawn from the study of electrospun nanofibers indicate that multiple parameters significantly impact the morphology and quality of the nanofibers. First off, high voltage during electrospinning has a major effect on nanofiber diameter; SEM study shows that the higher voltage produces finer fibers because it enhances polymer stretching. Second, the shape of the nanofibers is greatly influenced by changes in the flow rate of the precursor solution; greater flow rates result in bigger fiber diameters and possibly even bead formation. Furthermore, the distance between the needle tip and collector is critical since longer distances are associated with smaller fibers, which can be attributed to variables like solvent evaporation rate and deposition duration. Moreover, the diameter of the needle influences the morphology of the fiber; larger needles result in higher fiber diameters because they produce larger droplets with a lower surface tension. Furthermore, the diameter of the nanofibers is influenced by the solution concentration; higher concentrations result in larger fibers because of higher solution viscosity. Lastly, nanofiber diameter is influenced by both solution composition and voltage change; finer fibers are produced at higher voltages. Comprehending and refining these parameters is crucial in customizing nanofiber attributes for particular uses and refining electrospinning procedures to ensure effective nanofiber generation. A significant relationship has been found between the nanofiber diameter, wettability, and porosity. The wettability of nanofibers is a crucial factor in determining the absorption of proteins and the adherence of cells in biomedical applications. Wettability was evaluated by measuring water contact angles, which showed important trends. A decrease in hydrophilicity was indicated by an increase in water contact angles with increasing chitosan nanofiber concentrations (2%, 6%, and 8% wt%). On the other hand, water contact angles dropped at higher voltages (14, 16, 18, and 20 kV), indicating increased hydrophobicity. Higher flow rates and different needle diameters did not change this tendency, suggesting a relationship between voltage, tip-to-target distance, and hydrophobicity of the nanofiber. Furthermore, our study of nanofiber membranes showed that porosity levels consistently exceeded 30%, which is directly related to the wettability and diameter of the nanofibers. Porosity was difficult to control since fiber diameter and porosity are closely related to the electrospinning parameters. The porosity levels were found to be altered by changes in flow rate, voltage, salt addition, needle diameter, and solution concentration. This indicates the intricate interaction of parameters involved in maximizing membrane qualities for a range of applications. Based on the results, the electrospun chitosan nanofiber mats showed effective antibacterial activity toward both gram-positive and gram-negative bacteria. This work provides a basic understanding of the design of an efficient nanofiber-based antibacterial wound dressing material. Furthermore, chitosan nanofibers serve as a promising potential antibacterial agent.

Chitosan nanofibers, made via electrospinning, show promise for wound dressings. Their advantages include built-in antibacterial activity, high surface area for better bacterial interaction, and promotion of wound healing. They are biocompatible and biodegradable too. However, limitations include potential limitations in broad-spectrum antibacterial activity, a delicate structure requiring reinforcement, and higher production costs. Chitosan’s acidity might also irritate some wounds, and consistent nanofiber properties require careful processing control.

## Figures and Tables

**Figure 1 polymers-16-01984-f001:**
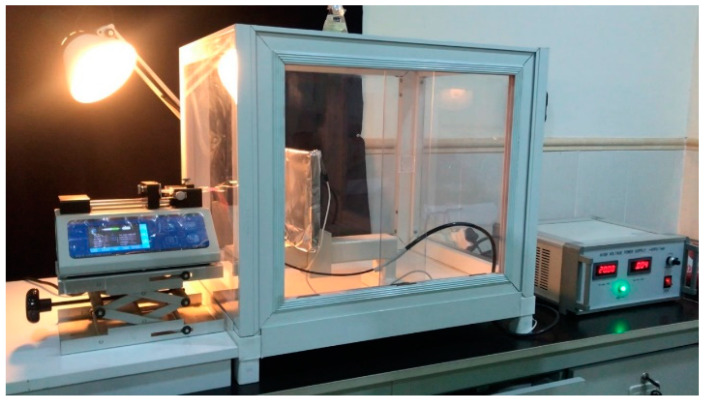
Electrospinning system.

**Figure 2 polymers-16-01984-f002:**
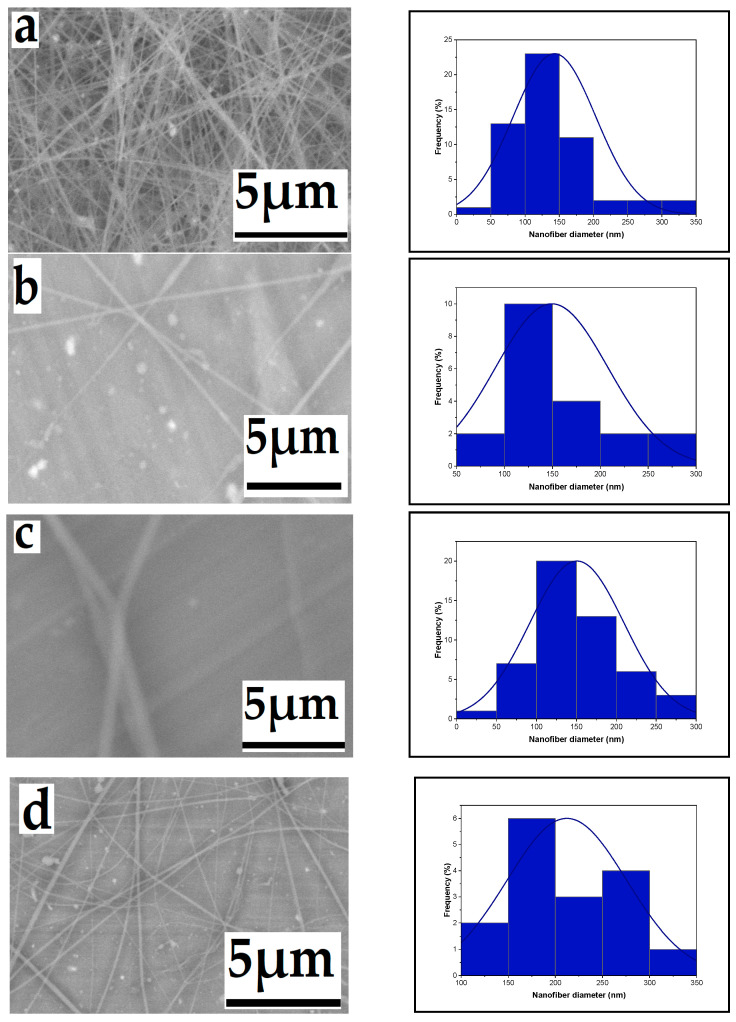
Effect of high voltage on the average nanofiber diameter distribution ((**a**): 143, (**b**): 149, (**c**): 160, and (**d**): 212 nm), respectively, from a 6% wt chitosan/TFA solution (tip-to-target distance: 10 cm; flow rate: 0.4 mL/h), high voltage: 20, 18, 16, and 14 kV.

**Figure 3 polymers-16-01984-f003:**
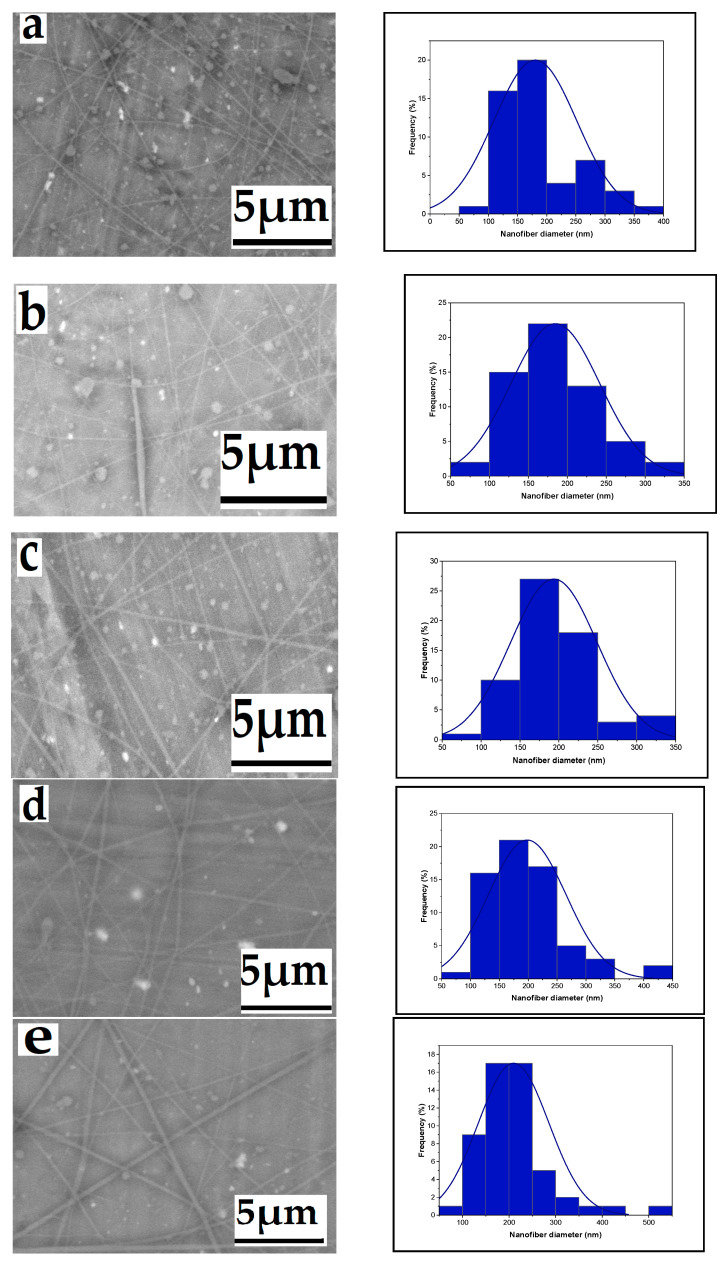
Effect of flow rates on the average nanofiber diameter distribution ((**a**): 180, (**b**): 184, (**c**): 193, (**d**): 198, and (**e**): 209 nm), respectively, from a 6% wt chitosan/TFA solution (high voltage: 20 kV; tip-to-target Distance: 10 cm), flow rate: 0.1, 0.2, 0.3, 0.4, and 0.5 mL/h.

**Figure 4 polymers-16-01984-f004:**
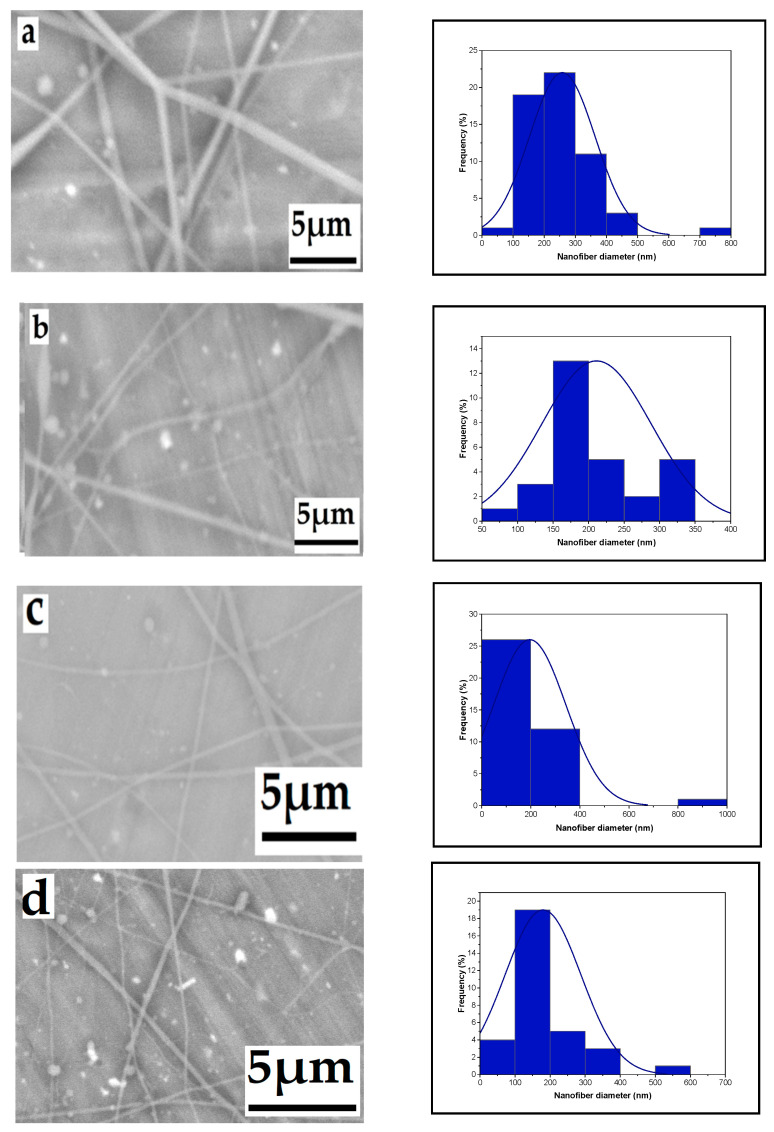
Effect of tip-to-target distance on the average nanofiber diameter distribution ((**a**): 258, (**b**): 2556, (**c**): 228, and (**d**): 179 nm), respectively, from a 6% wt chitosan/TFA solution (high voltage: 20 kV; flow rate: 0.4 mL/h), distance: 12, 14, 16, and 18 cm.

**Figure 5 polymers-16-01984-f005:**
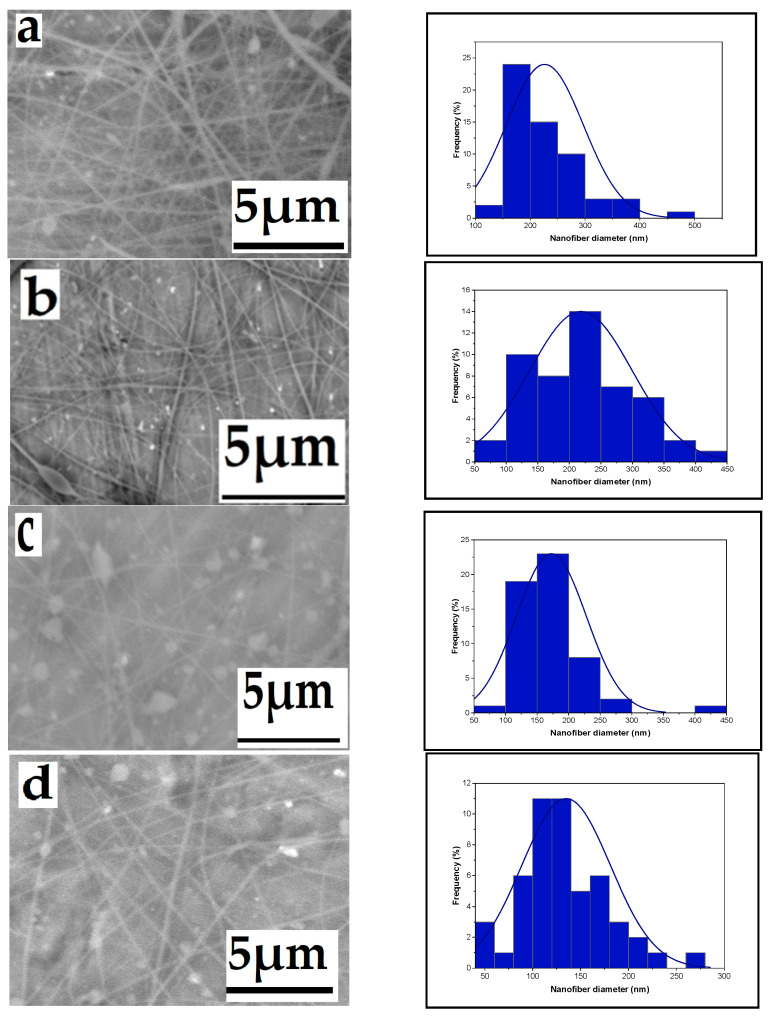
Effect of needle diameter on the average nanofiber diameter distribution ((**a**): 226, (**b**): 218, (**c**): 172, and (**d**): 134 nm), respectively, from a 6% wt chitosan /TFA solution (high voltage: 20 kV; flow rate: 0.4 mL/h, tip-to-target distance: 10 cm), needle diameter: 0.6, 0.5, 0.4, and 0.3 mm.

**Figure 6 polymers-16-01984-f006:**
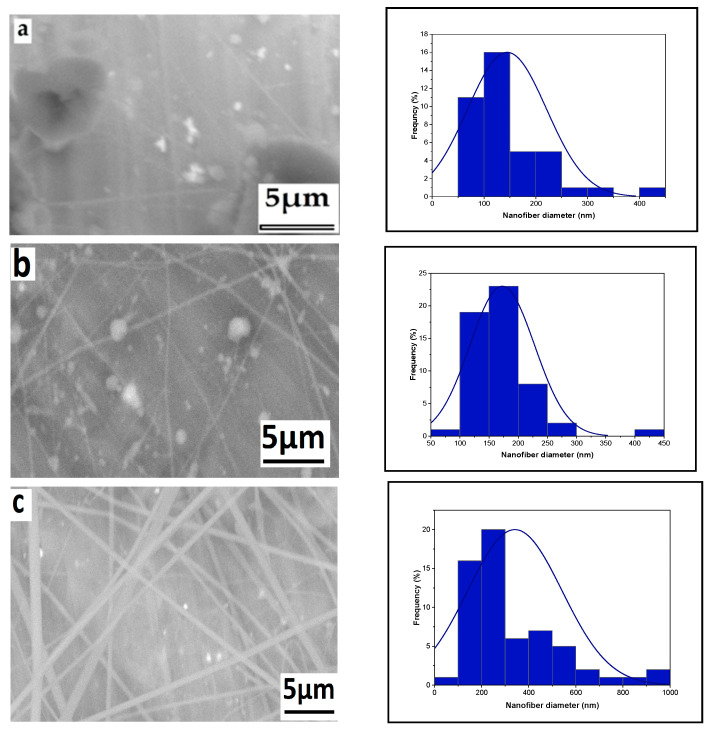
Effect of solution concentration on the average nanofiber diameter distribution ((**a**): 144, (**b**): 172, and (**c**): 367 nm), respectively, from a 2% wt, 6% wt, and 8% wt chitosan/TFA solution (voltage: 20 kV; flow rate: 0.4 mL/h; tip-to-target distance: 10 cm; needle diameter: 0.6 mm).

**Figure 7 polymers-16-01984-f007:**
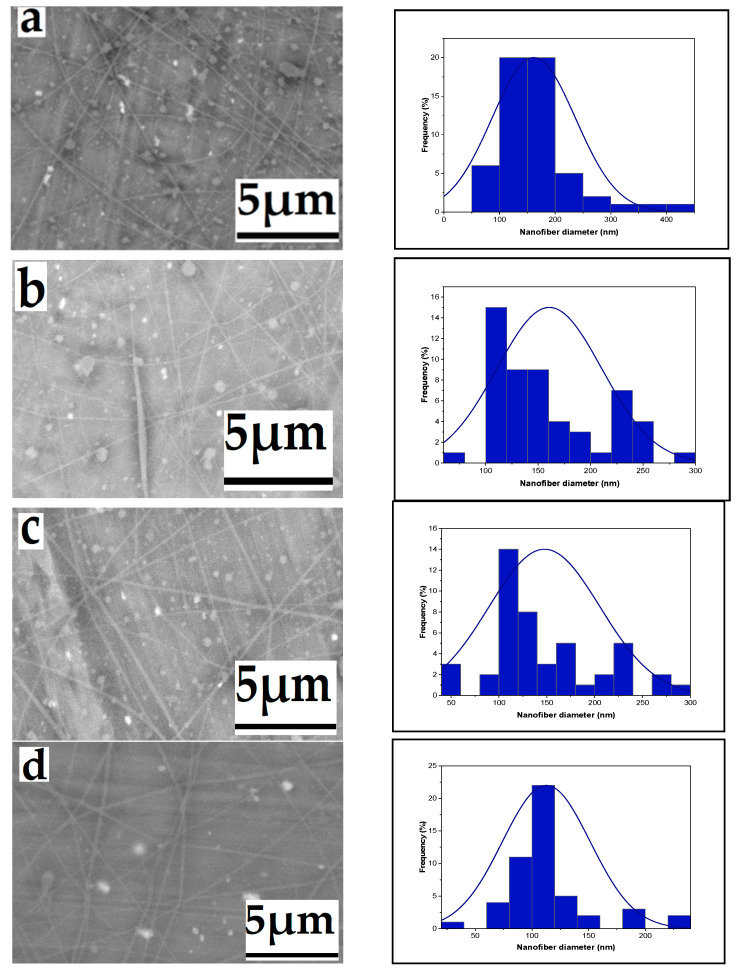
Effect of high voltage with salt on the average nanofiber diameter distribution ((**a**): 161, (**b**): 160, (**c**): 147, and (**d**): 112 nm), respectively, from a 6% wt chitosan/TFA solution (tip-to-target distance: 10 cm; flow rate: 0.4 mL/h), high voltage: 20, 18, 16, and 14 kV.

**Figure 8 polymers-16-01984-f008:**
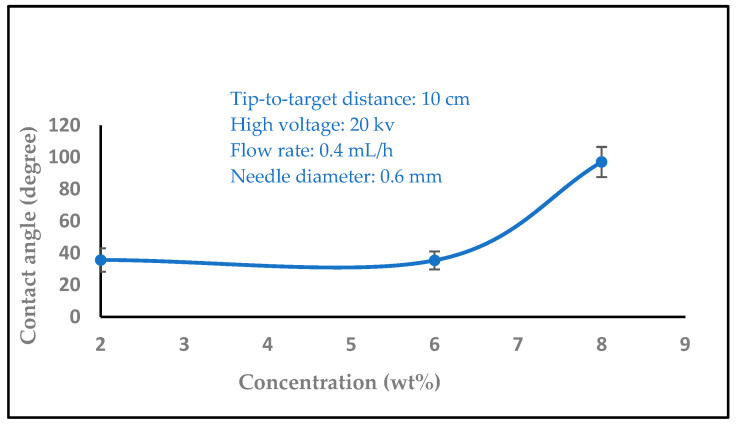
Variation of the contact angle of water droplets on electrospun chitosan nanofiber mat with the solution concentration.

**Figure 9 polymers-16-01984-f009:**
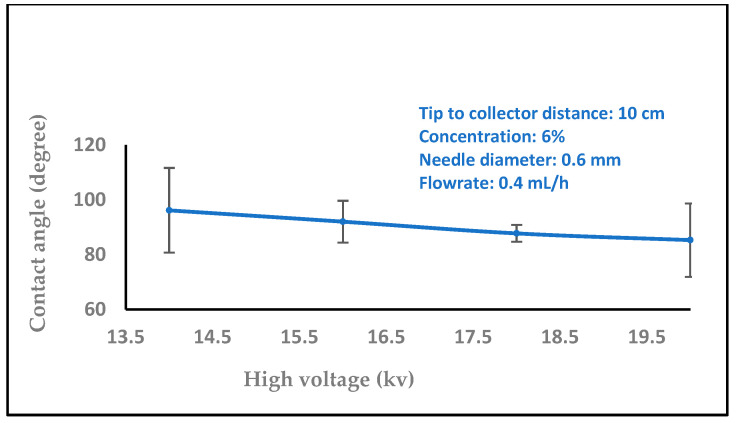
Variation of the contact angle of water droplets on electrospun chitosan nanofiber mat with the high voltage.

**Figure 10 polymers-16-01984-f010:**
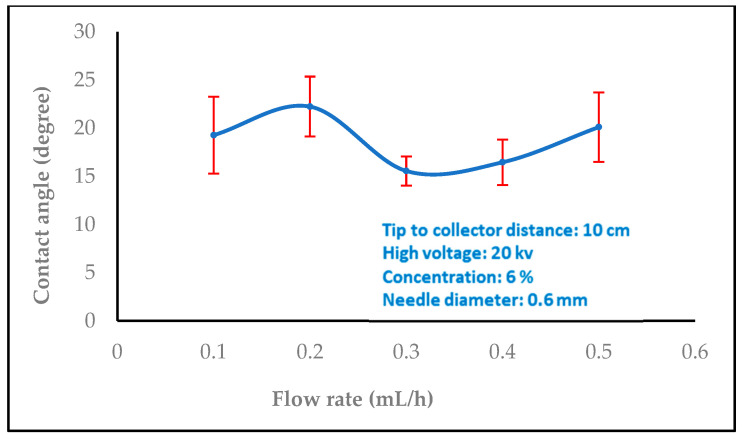
Variation of the contact angle of water droplets on electrospun chitosan nanofiber mat with the flow rate.

**Figure 11 polymers-16-01984-f011:**
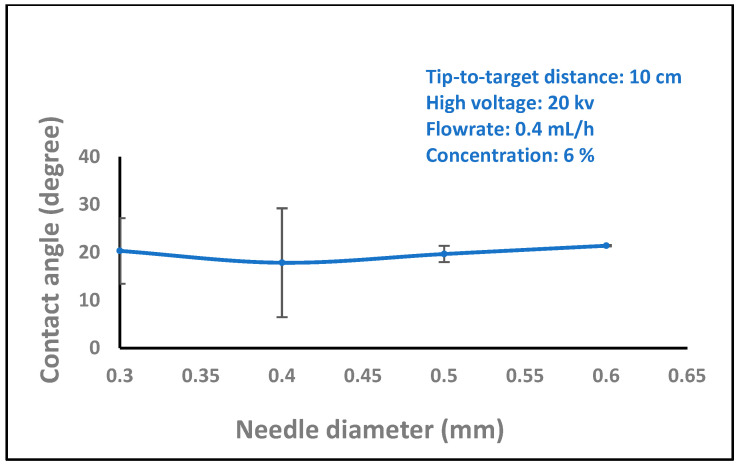
Variation of the contact angle of water droplets on electrospun chitosan nanofiber mat with the needle diameter.

**Figure 12 polymers-16-01984-f012:**
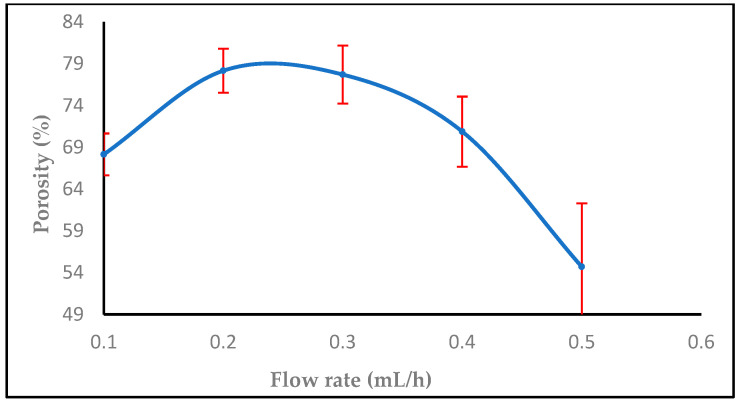
The effect of increasing flow rate on the porosity (%).

**Figure 13 polymers-16-01984-f013:**
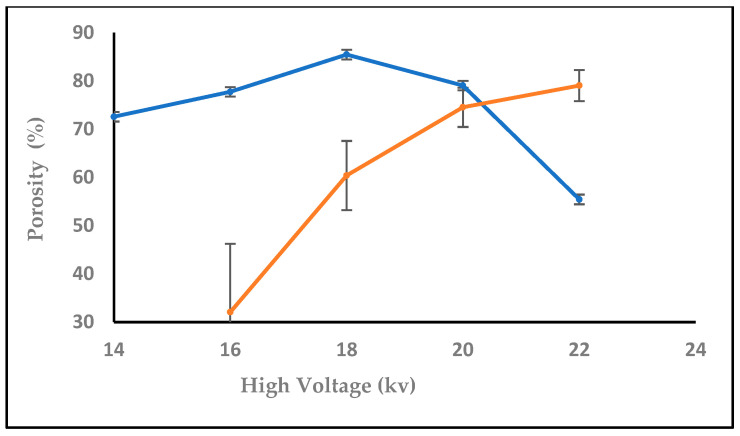
The effect of increasing high voltage (brown line) and high voltage with adding salt (blue line) on the porosity (%).

**Figure 14 polymers-16-01984-f014:**
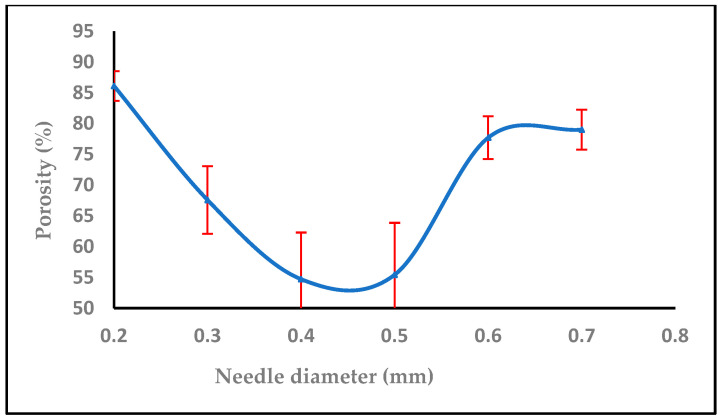
The effect of increasing needle diameter on the porosity (%).

**Figure 15 polymers-16-01984-f015:**
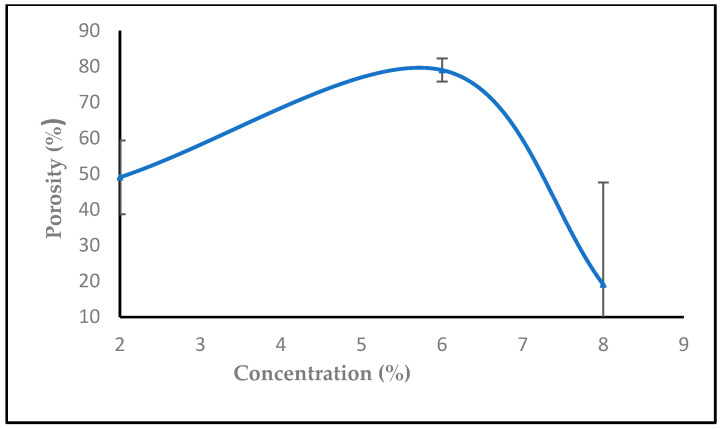
The effect of increasing concentration on the porosity (%).

**Figure 16 polymers-16-01984-f016:**
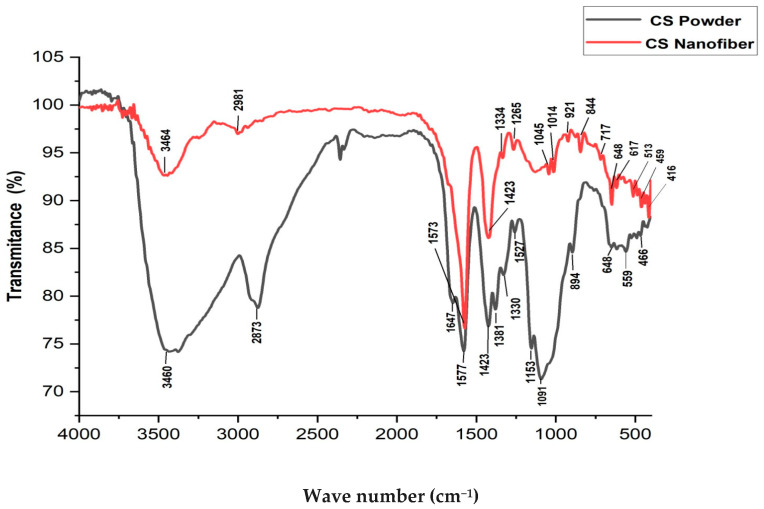
FTIR spectra of the chitosan powder (black) and chitosan nanofiber (red).

**Figure 17 polymers-16-01984-f017:**
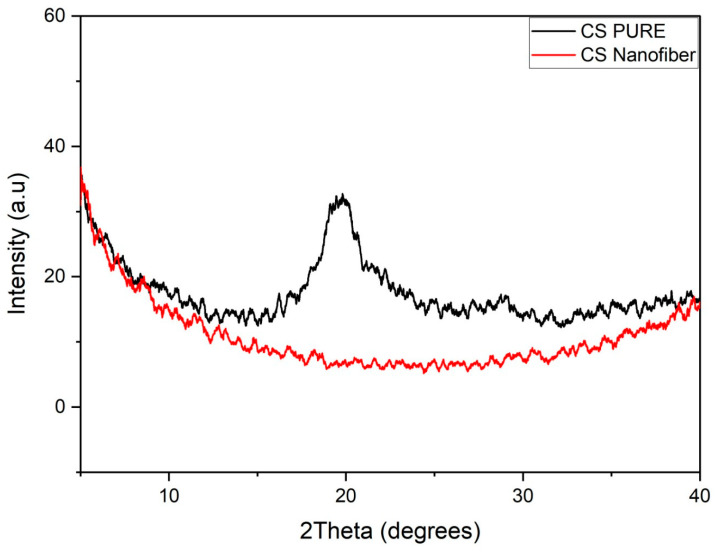
XRD patterns of chitosan polymer (CS) and chitosan nanofiber (CS nanofiber).

**Figure 18 polymers-16-01984-f018:**
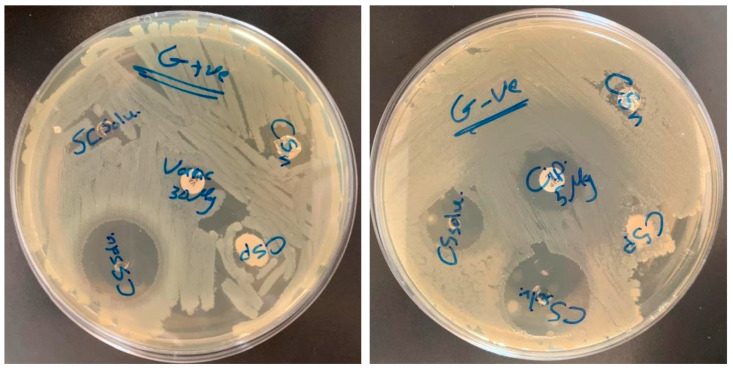
Antibacterial activity against Gram positive bacteria (*S. aureus*) (**Right**) and Gram negative bacteria (*E. coli*) (**Left**) for chitosan polymer (CSp) as well as chitosan nanofiber (CSn).

**Table 1 polymers-16-01984-t001:** Antibacterial test measurements.

Material	*S. areous*		*E. coli*
chitosan powder (CSP)	8 mm	chitosan powder (CSP)	7 mm
chitosan nanofiber (CSN)	18 mm	chitosan nanofiber (CSN)	11 mm

## Data Availability

All data are included within the article.

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
