# Peer review of "Characterization of Electrospinning Chitosan Nanofibers Used for Wound Dressing"

_polymers, 2024, doi:10.3390/polym16141984_

Round 1
Reviewer 1 Report
Comments and Suggestions for Authors
1. Please describe the antibacterial experiment in the section of Materials and Methods.
2. Please discuss Figure 18. Why did chitosan polymer show no antibacterial activity against E. Coli and S. aureus? This result is different from the published literatures. Authors may provide a table, showing the change of cells numbers of E. Coli and S. aureus during this experiment.
3. Please reorganize paragraphs in the introduction section, for example, put paragraphs 1 and 2 into one.
4. What kind of antibacterial and bioactive substances were incorporated into chitosan nanofiber? Several?
5. Please describe/show chemical structure of the studied chitosan polymer.
6. It is suggested to study wound healing properties.
7. There are 18 figures in this manuscript, and authors may put some figures into one figure. 8. 3.1.4. should be ‘Effect of Needle Diameter’.
9. Please revise the text and references format in the manuscript according to journal style.
10. Please check the text in the conclusions.
Comments on the Quality of English LanguageAuthors need revise all text and references according to journal style.
Author Response
Dear Academic Editor and Reviewers,
Thank you sincerely for your letter and for forwarding the reviewers' insightful comments regarding our manuscript titled” Characterization of Electrospinning Chitosan Nanofibers Used for Wound Dressing” (ID: polymers-2972452).
We deeply appreciate the constructive feedback provided, which we find immensely valuable for refining and enhancing our paper
The guidance offered by the reviewers is instrumental not only in improving this manuscript but also in shaping the trajectory of our ongoing research endeavors.
We have carefully examined the reviewers' comments and have seriously incorporated the suggested revisions into the manuscript. These revisions have been appropriately highlighted within the revised document. We are optimistic that our revisions address the concerns raised and align with the standards expected by the journal.
Once again, we extend our gratitude for the invaluable feedback and the opportunity to improve our work. We eagerly await further guidance from you and the reviewers.

Reviewer 2 Report
Comments and Suggestions for Authors
The manuscript “Characterization of Electrospinning Chitosan Nanofibers Used for Wound Dressing” describes the preparation of chitosan nanofibers for wound dressing applications. The research outlined a systematic study of different operational parameters and their influence upon the morphology, wettability, and porosity of electrospun chitosan nanofibers. Also, the authors conducted a series of characterizations to evaluate the properties and efficiency of the obtained nanofibers against both Gram-positive and Gram-negative microorganisms.
The results are promising for practical applications. The reported work is quite interesting, conducted and presented well; however, the manuscript needs a revision before its publication. Some remarks need to be addressed:
1) It is important to provide a broader context for the problem of Wound dressings in promoting wound healing and justify the need to develop new materials. The authors are encouraged to point out the current limitations of existing methods and highlight the advantages of a combined approach using chitosan nanofibers. A comparison of the proposed technology with existing methods would help to evaluate its advantages and competitiveness.
2) The chapter on conclusions might use some improvement, in my opinion. Please reword the Conclusion part and introduce more specific results. Analyze the results obtained and compare them with previous research in this field. Discuss the advantages and disadvantages of the developed method.
Comments on the Quality of English LanguageMinor editing of English language is required.
Author Response

(The authors gave the same response as above.)

Round 2
Reviewer 1 Report
Comments and Suggestions for Authors
1. lines 395 and 416,should be the peak at 3460 cm⁻¹, not 4360.
2. "The peaks at lower wavenumbers, such as 648 cm⁻¹, 559 cm⁻¹, and 466 cm⁻¹, are attributed to the skeletal vibrations of glycosidic bond, characteristic of chitosan. " It is incorrect, please delete it. C-O-C stretching vibrations correspond to glycosidic bonds.
3. lines 458 and 466, please put references on the previous studies.
Comments on the Quality of English LanguageExtensive editing of English language is required
Author Response
Dear Reviewer,
Thank you for your valuable feedback and suggestions on our manuscript. We have carefully considered your comments and made the necessary revisions accordingly.
In the second round of review, you requested minor revisions, which we have addressed in detail. Additionally, you recommended improving the introduction and conducting extensive editing of the English language. We have made significant improvements to the introduction to better frame the context and objectives of our study. Furthermore, we have thoroughly edited the manuscript for clarity, coherence, and correctness in the English language.
We believe these changes have strengthened the overall quality of our manuscript, and we are grateful for your insightful guidance. We hope the revised version meets your approval.
Thank you once again for your time and consideration.
Best regards,
Manaf A. Mahammed

Reviewer 2 Report
Comments and Suggestions for Authors
The paper is significantly improved and can be published in this form!
Author Response
Reviewer 2 did not send comments in the second round of revision.